# A Formative Assessment of Antibiotic Dispensing/Prescribing Practices and Knowledge and Perceptions of Antimicrobial Resistance (AMR) among Healthcare Workers in Lahore Pakistan

**DOI:** 10.3390/antibiotics11101418

**Published:** 2022-10-16

**Authors:** Noor Sabah Rakhshani, Linda Marie Kaljee, Mohammad Imran Khan, Tyler Prentiss, Ali Turab, Ali Mustafa, Memoona Khalid, Marcus Zervos

**Affiliations:** 1PHC Global (pvt.) Ltd., 241 Bahadur Shah Zafar Road, Bahadurabad 3, Karachi 74800, Pakistan; 2Global Health Initiative, Henry Ford Health, One Ford Place, Suite 1E, Detroit, MI 48202, USA; 3Division of Infectious Diseases, Henry Ford Health, Clara Ford Pavillion, Detroit, MI 48202, USA

**Keywords:** antimicrobial resistance, AMR stewardship, dispensing practices, community health care, Pakistan, South Asia

## Abstract

(1) Background: Antimicrobial resistance causes significant disease burden in low- and middle-income countries. The objective of this paper is to describe antibiotic dispensing/prescribing practices and underlying factors associated with these practices among community-based healthcare workers. (2) Methods: Cross-sectional survey data were collected from private and public health facilities in 14 union councils, Lahore Pakistan. Respondents included physicians, nurses, lady health workers/volunteers, midwives, pharmacy and medicine shop employees, and medical technicians. Descriptive and bivariate analysis are used to present the data; (3) Results: 177 respondents completed the survey. In terms of weekly dispensing of antibiotics, the most common were Amoxicillin/Augmentin (2.3 [SD 1.5]), Cefixine (2.4 [SD 1.6]), and Azithromycin (2.5 [SD 2.1]). For children, antibiotics were more likely to be prescribed/dispensed for sore throat (54.3%/95) and diarrhea (48.9%/86). For adults, antibiotics were more likely to be prescribed/dispensed for sore throat (67.0%/118), diarrhea (59.7%/105) and burning sensation when urinating (55.7%/176). In total, 55.4% of respondents stated that they have sold partial antibiotic courses to patients/customers. A total of 44.6% of respondents incorrectly answered that antibiotics could be used for viral infections; (4) Conclusions: Data from this study and similar research emphasize the urgent need to implement community-based stewardship programs for all healthcare workers.

## 1. Introduction

Global data suggest that a majority (~85% to 95%) of antibiotic dispensing and utilization occurs within communities [1,2]. Antibiotic usage has seen exponential increase in low- and middle-income countries (LMIC) over the last decade with antimicrobial resistance (AMR) disproportionately impacting low-resource settings [3,4] Many factors contribute to misuses of antibiotics including availability of cheaper generic pharmaceuticals and unregulated over-the-counter pharmacy dispensation or unenforced prescribing policies. In addition, many health care facilities in LMIC have limited laboratory facilities to support testing to identify bacterial resistance and susceptibility and enable definitive prescribing practices [5,6].

In Pakistan, antibiotics are classified as prescription-only. However, these regulations are not stringently enforced due in part to shortages of a qualified workforce in retail pharmacies and public-sector hospitals, as well as inadequate licensed healthcare providers in rural areas. The World Health Organization (WHO) recommends one pharmacist per 2000 population. Pakistan has 0.9 pharmacists per 100,000 population [7,8]. However, there are more than 76,000–88,000 registered drugs in Pakistan and more than 80,000 pharmacies and medicine shops of which only 20% are licensed [7]. The average number of drugs prescribed to patients is greater than three per consultation with a physician and an estimated 50% of the population takes prescription-only drugs without consultation to off-set the cost of health service providers. The rate of antibiotic misuse is alarming and widely practiced both in public and private health sectors [9]. 

From a cluster survey in Sindh Province, among children <5 years living in Karachi presenting to a health facility with fever in the past two weeks, 27% were prescribed an oral antibiotic. Similarly, among children presenting with diarrhea, 10% received an oral antibiotic and 6% an injection [10]. While some of these prescriptions were likely appropriate, these data indicate the need for more information regarding how and when antibiotics are dispensed/prescribed in community health facilities. In a study in Islamabad and Rawalpindi Pakistan, 386 clients attending one of four pharmacies received 525 courses of antibiotics. Approximately 35% of the study participants received antibiotics without a prescription [11].

Misuse and over prescribing and dispensing of antibiotics is not unique to Pakistan. In a study in Kathmandu Nepal, 36.9% to 67.6% of community pharmacists stated they would ‘mostly likely’ or likely’ dispense antibiotics to adult patients with sore throat and diarrhea, respectively. The proportions for pediatric patients were even higher at 62.2% for sore throat and 80.2% for cough or diarrhea [12]. Utilizing a medical audit database in India, an estimated 412 antibiotics/1000 persons were dispensed in a one-year period. The age group most likely to receive antibiotics was 0–4 years [13]. 

The World Health Organization South-East Asia (SEA) region has the highest risk of emerging drug resistance and spread of resistant pathogens. Megacities such as Lahore Pakistan are particularly susceptible due to large reservoirs of drug resistant pathogens, inadequate water and sanitation infrastructures, large and densely populated slums, and on-going rural to urban migration [14]. While there is limited data on multi-drug resistance in Pakistan, a recent review of 55 studies indicates an overall pooled proportion of Extended Spectrum beta-Lactamase (ESBL) producers at 0.40% (95% CI: 0.34–0.47) and overall significant heterogeneity [15]. In a clinical study with chronic renal patients, the prevalence of ESBL-producers was 100% in *K. pneumoniae* and 75% in *E. coli*. Carbapenem resistance was evident in 79% of *K. pneumoniae* and 5.4% of *E. coli* [15]. Substantial scientific literature has documented the emergence of extensively drug resistant (XDR) *S.* Typhi in Pakistan. Between November 2016 and August 2021, a total of 21,124 (XDR) *S.* Typhi have been reported in Sindh Province. This outbreak has been declared the largest drug-resistant typhoid outbreak to date. Isolates have shown an array of genetic modifications with resistance to multiple first-line antibiotics (chloramphenicol, ampicillin, trimethoprim-sulfamethoxazole) as well as fluoroquinolones and third-general cephalosporins [16]. 

At the policy level, Pakistan crafted their AMR National Action Plan following the November 2016 outbreak of XDR *S.* Typhi. The rationale for the plan was to ensure that policies are in place for providers and the health management cadre to optimize use of antibiotics and establish a monitoring system to assess antibiotic consumption and use. Pakistan also formed its international alliance coupled with the development of the Pakistan Global Antibiotic Resistance Partnership (GARP) in 2016. However, translation of policies into enforced AMR stewardship programs to decrease rampant prescribing, dispensing, and use of antibiotics has not occurred. 

For AMR stewardships to be successful, they must be context-specific and address social and economic barriers at the policy, programmatic, and individual levels [17,18,19]. In this paper, we present cross-sectional data from Lahore Pakistan community health providers including physicians, nurses, lady health workers/volunteers, midwives, pharmacists, medicine shop employees, and medical technicians. Medical technicians refer to allied health professionals that support other healthcare workers but may also provide some patient care.

The primary objective of this paper is to assess and describe antibiotic dispensing/prescribing practices and underlying factors associated with these practices among community-based healthcare workers. These data will be used to determine targeted populations and feasible approaches to community-based stewardship programs in Lahore.

## 2. Results

### 2.1. Demographics

177 respondents completed the survey. Overall, 53.1% (94) of respondents were female and mean respondent age was 32.4 (SD 10.0) years. Age range was 19 to 64 years. A majority of physicians and nurses were female, and a majority of pharmacists and technicians were male. All medicine shop owners and compounders were male. Most of the respondents were from urban settings (93.2%/164) (Table 1).

### 2.2. Healthcare Facilities

Number of persons employed varied considerably both within and across types of health facilities. The mean number of employees at the health care facilities ranged from 6 [range 3 to 15] at the basic health units (BHUs) to 154 [range 5 to 350] at the referral hospitals. The private clinics had an average of 18 employees [range 1 to 50] and the pharmacies and medical shops had 7 [range 1 to 22]. In terms of weekly mean number of patients, the private clinics had the fewest (209 [range 4 to 1260]) and the pharmacies and medicine shops had the most (1265 [range 100 to 4200]). The BHUs had an average of 370 patients [range 30 to 800] and the referral hospitals had an average of 1164 [range 36 to 15,000]. Overall, 35.6% (63) of respondents were from pharmacies and medicine shops, 34.5% (61) from primary care centers, and 11.9% (21) and 18.1% (32) from secondary and tertiary care facilities.

Respondents were asked to rank common infections seen within their facilities. Overall, upper respiratory and gastrointestinal infections were the most common across health care facilities. However, there were significant differences in reported common infections between the four types of facilities. Urinary tract infections were more common in private clinics and hospitals. Skin and wound infections were more common in BHUs and pharmacies and medicine shops (Table 2).

### 2.3. Antibiotic Dispensing and Prescribing

Non-physician respondents were asked percentage of patients or customers requesting antibiotics without prescription. Physicians were asked about percentage of patients asking for antibiotics. There was no significant difference by type of health care worker (*p* = 0.66). However, those reporting the highest rates for >50% patient/customer requesting antibiotics were lady health workers/volunteers (40.0%/4), medicine shop owners and compounders (46.2%/6), and medical technicians (50.0%/3). Over 38% (15) of physicians reported that >50% of their patients request antibiotics. Overall, 55.4% of respondents stated that they have sold partial antibiotic courses to clients or patients without enough money to buy the entire course. However, there were significant differences across healthcare professions with 97.9% (47) of pharmacists and 93.8% (15) of medicine shop owners and compounders stating they have sold partial antibiotic courses to their clients compared to physicians (22.2%/6), nurses (7.1%/2), and midwives (28.6%/2) [*p* < 0.001].

Overall, in terms of regular (weekly) dispensing of specific antibiotics, the most common were Amoxicillin/Augmentin (2.3 [SD 1.5]), Cefixine (2.4 [SD 1.6]), and Azithromycin (2.5 [SD 2.1]). Least common were Co-trimoxazole (4.2 [2.6]) and Ampicillin (3.8 [SD 2.6]). However, for all seven antibiotics there were significant differences by type of health care worker. Nurses, lady health workers/volunteers, and midwives were much more likely to report regular use of Ceftriaxone, which is an injectable antibiotic (Table 3).

Overall, for children, antibiotics were more likely to be prescribed/dispensed for sore throat (54.3%/95) and diarrhea lasting more than one day (48.9%/86). However, there were significant differences between health workers in terms of likelihood of prescribing/dispensing antibiotics for fever, sore throat, and diarrhea (Table 4). Overall, for adults, antibiotics were more likely to be prescribed/dispensed for sore throat (67.0%/118), diarrhea (59.7%/105) and burning sensation when urinating (55.7%/176) (Table 5). Across four symptoms (fever, cough, sore throat, and diarrhea), antibiotics were significantly more likely prescribed/dispensed for adults (*p* < 0.001). In terms of wound care, antibiotics were highly likely to be prescribed across all types of healthcare workers though there were significant differences. In terms of both redness and/or swelling around a wound and wound discharge, 100% of the lady health workers/volunteers reported that they would dispense an antibiotic (Table 6).

### 2.4. Familiarity with Guidelines and Policies

Overall, 32.0% (56) of respondents stated that they were familiar with the World Health Organization AWaRe (access, watch, reserve) categories related to recommended use of antibiotics. Physicians (55.0%/22) were most familiar with the guidelines and pharmacy manager and clerks (18.8%/9) and medical technicians (14.3%/1) were least familiar (*p* = 0.014). There was more familiarity across types of health workers in regard to antibiotic prescribing/dispensing guidelines from the Punjab Healthcare Commission. Over 50% of physicians, nurses, Lady Health Workers/Volunteers and midwives reported knowledge of these guidelines. Fewer pharmacists (24.5%/12), medicine shop owners and compounders (39.1%/9), and medical technicians (28.6%/2) were familiar with the local guidelines.

All the pharmacists and medical shop owners and compounders reported that their shop was registered as a pharmacy with the Punjab Provincial Healthcare Commission, Primary and Secondary Healthcare Department, Chief Drug Controllers Office. In addition, 79.6% (39) of pharmacists and 76.9% (10) medicine shop owners and compounders reported having a digital record-keeping system to monitor antibiotic sales.

### 2.5. Knowledge and Perceptions of AMR

There were significant differences between types of health workers in terms of familiarity with antimicrobial resistance. Respondents were asked if they had “heard of antibiotic resistance”. One hundred percent of physicians stated ‘yes’; however, only 28.6% (2) of medical technicians, 59.2% (29) of pharmacists, 65.2% (15) of medicine shop owners and compounders, and 65.5% of nurses had heard of antibiotic resistance. Over 72% (13) of midwives and 80.0% (8) of lady health workers/volunteers stated they had heard of antibiotic resistance. On a 12-point knowledge scale, the overall mean score was 8.4 (SD (1.7)) [range 5 to 12]. Overall, 44.6% of respondents incorrectly answered that antibiotics could be used for viral infections, 49.2% incorrectly answered that antibiotics could be used to cure colds and flu, and 51.1% incorrected answered that antibiotics could be used to treat COVID-19. In terms of the item on COVID-19, 36.6% of physicians to 60.9% of medicine shop owners and compounders incorrectly answered the question.

The overall mean score for perception of the severity of AMR was 12.0 (SD 1.7) [range 8 to 16] and for self-efficacy to make changes in dispensing/prescribing practices the mean score was 9.0 (SD 1.2) [range 6 to 12]. The overall mean score for response efficacy related to changing antibiotic dispensing/prescribing practices was 5.5 (SD 0.9) [range 3 to 8] and for response costs associated with changing practices was 4.6 (SD 1.2) [range 1 to 7] (Table 7).

## 3. Discussion

In 2019, an estimated 4.95 million deaths were associated with bacterial resistance of which 1.27 million deaths were directly attributable. The highest burden of AMR is in Sub-Saharan Africa and South Asia [20]. Data from the United States indicate that AMR burden has increased during the COVID-19 pandemic due to multiple factors including long-term hospitalization, resource reallocation to address the pandemic, and use of antibiotics to treat febrile and pulmonary symptoms associated with COVID-19 [21]. While bacterial co-infections in COVID-19 patients may require use of antibiotics, a study of hospitalized patients in Spain indicates that over 87% were prescribed antibiotics despite relatively low rates of definitive co-infection [22]. In addition, a study in France examined self-medication during COVID-19 including use of antibiotics and related adverse drug reactions [23]. Limited data are available on the impact of COVID-19 on AMR in LMIC; however, it is anticipated that burden of bacterial resistance pathogens has increased over the past two years [24]. In a study from Colombia, azithromycin was one of the most dispensed medications at drug stores and pharmacies for treatment of COVID-19 [25]. WHO research from late 2020 indicates that 35/56 (63%) of countries studied showed an increase in prescriptions of antibiotics 13/35 (37%) reported increase in multi-drug resistant organisms [26]. In our data from Lahore in 2021, we see that over 50% of healthcare workers believed that antibiotics were a treatment for COVID-19. This suggests the possibility that dispensing and prescribing of antibiotics increased particularly in community settings. Antimicrobial stewardship needs to address optimizing use of antibiotics when indicated for COVID-19 patients and education and training to decrease misuse of antibiotics during the pandemic [27].

Overall, our data indicate that there is room for improvement in terms of knowledge about AMR across all health disciplines including physicians. However, knowledge does not necessarily translate into practice and therefore it is essential that community healthcare workers have the needed skills and resources to support community stewardship (i.e., communication with patients/clients, infection prevention and control, antibiotic guidelines, information on alternative over-the-counter pharmaceuticals for common viral disease) [28].

The World Health Organization has developed multiple tools for use in LMIC health settings to support healthcare workers in establishing AMR stewardship programs and practices [19,29,30]. In addition, the WHO AWaRe guidelines provide an up-to-date list of antibiotics by three categories—access, watch, reserve [31]. In our data, across healthcare professions, azithromycin, cefixime, and ciprofloxacin were ranked highly in terms of weekly prescribing/dispensing of antibiotics. These three antibiotics are under the ‘watch’ category defined as “highest-priority critically important antimicrobials…recommended only for specific, limited indications” [32]. In addition, nurses and lady health workers/volunteers reported common use of ceftriaxone which is in the ‘watch’ category and is delivered through injection. These findings mirror other research in Pakistan. In a study of World Health Organization essential drugs in four regions of Lahore, azithromycin and ciprofloxacin were readily available in all sixteen participating private pharmacies [33]. In another research project in Lahore with 353 pharmacies and medical stores, 97% dispensed antibiotics without a prescription to “standardized patients”. The most frequent antibiotic dispensed was ciprofloxacin [34].

A large proportion of all respondents stated they would likely or very likely prescribe/dispense antibiotics for fever, cough, sore throat, diarrhea, burning sensation during urination, and wound care. For the first four symptoms, healthcare providers were significantly more likely to prescribe/dispense antibiotics for adults. This is different than in Nepal where pharmacists were more likely to prescribe/dispense antibiotics for children for these same symptoms [12].

Pharmacies and medicine shops are frequently first stops for persons who are sick in LMIC. This is particularly true in rural areas and urban slums where other health resources are limited or non-existent. However, data indicate that pharmacy and medicine shop personnel not only dispense antibiotics without prescriptions, but also sell shorter courses than recommended and have inadequate referral processes for clinical care [35,36]. In the current study, over 97% of pharmacy respondents and over 93% of medicine shop respondents stated that they have sold partial antibiotic courses to clients.

Data from this study and other similar research emphasizes the urgent need to decrease antibiotic prescribing and dispensing. Physicians scored highest on perceived severity of AMR, self-efficacy for being able to change practices, and response efficacy in terms of understanding how practice changes can contribute to decreasing AMR. Pharmacists and medicine shop personnel scored highest on the response cost scale which measures concerns about negative social-economic outcomes from decreasing dispensing of antibiotics. These data suggest that different approaches and emphasis are necessary to address concerns and barriers to stewardship among some community healthcare workers while supporting facilitators to stewardship among other providers.

Community AMR stewardship requires integrated programs and policies. Education and training are imperative for all healthcare workers, as well as community members, to support stewardship. However, there is also a need to support infection prevention and control and immunization programs that can contribute to decreasing disease burden and use of antibiotics [20]. As AMR National Action Plans are established globally, policies must be enforced (e.g., prescription-only regulations) and translated into contextually relevant and feasible programs. Health facility and institutional leaders need to establish short- and long-term goals which can be accomplished within resource-limited settings and utilize materials, guidelines, and trainings which are available at the national and international levels [29].

The current study was conducted with 178 health care workers in 14 Union Councils within Lahore. The size of the study may limit generalization of the findings for elsewhere in Lahore and in Pakistan. However, our findings are consistent with and supportive of similar research in Pakistan and the region. The data on antibiotic prescribing and dispensing was subjective and based on reported practices. Further research is needed utilizing more objective methodologies including standardized patients and implementation of stringent antibiotic consumption and use protocols in healthcare facilities. The current data provide an overview of the magnitude of sub-optimal antibiotic prescribing and dispensing practices and potential areas of attention for targeted stewardship interventions.

## 4. Materials and Methods

### 4.1. Research Site

Lahore is the second largest city in Pakistan and is the capital of Punjab Province. According to the 2017 census, the population was over 11 million people with an annual growth rate of 4.1% since 1998 [37]. In 2022, the estimated population is 13 million. Lahore is divided into ten zones/towns, and each town is subdivided into union councils (UC). Between 45,000 and 125,000 people occupy a single UC. Over the last two decades, there has been a significant rural-to-urban migration throughout Pakistan and many UCs in Lahore are made up of ethnic groups from across the country. The current research was conducted in 14 UCs in Lahore and all respondents were of the same ethnic background (Figure 1). These UCs were selected based on geographic, socio-cultural, health, and economic factors including: (1) population density; (2) mean household income and primary occupations; (3) migration status (rural-to-urban); and (4) types of health services available including both public and private facilities. Prior to data collection, the UCs were mapped for locations of healthcare facilities and pharmacies and medicine shops.

### 4.2. Healthcare in Lahore

Many public healthcare facilities in Lahore provide heavily subsidized healthcare to the communities. These facilities are classified into five categories: basic health units (BHUs), rural health centers (RHCs), tehsil headquarter hospitals (THQs), district headquarter hospitals (DHQ), and teaching hospitals. Lahore also has a large private healthcare sector consisting of stand-alone clinics and tertiary level hospitals. All medical shops and many large pharmacies are also privately owned and only accept out of pocket payments

### 4.3. Research Design

The study followed a combined exploratory and explanatory mixed methods approach. The study included the following components: (1) key informant interviews with policy makers and program administrators; (2) qualitative interviews with community-based health care providers; and (3) a socio-economic cross-sectional survey of community-based health care providers. In this paper, data are presented from the cross-sectional survey which employed a purposive sampling methodology.

### 4.4. Research Populations

The cross-sectional survey was conducted with public and private health care workers including physicians, nurses, midwives, pharmacists, medicine shop owners, and medical technicians working in the 14 study UCs.

### 4.5. Survey Development

The survey instrument was adapted from a similar AMR and stewardship study conducted in Kathmandu, Nepal with physicians and pharmacists by one of the co-PIs [12]. Modifications were made to the survey to account for the socio-economic and health infrastructure in Lahore. The survey was also modified to address different issues for the broader target population in Pakistan. Survey sections included: demographics, infectious diseases among patients/clients, dispensing/prescribing practices, AMR knowledge, and perceptions of AMR severity, self-efficacy for change, response efficacy for changing dispensing/prescribing practices, and response costs for making changes (see Table 8 and Appendix A). An additional section on economic and social drivers of dispensing antibiotics was included but is not a part of the current analysis.

### 4.6. Pilot of the Survey Instrument

The survey was piloted prior to data collection. The pilot was with seven respondents including two physicians, a lady health volunteer, midwife, medical shop dispenser, and two pharmacists (hospital-based and private shop). The pilot supported field training for the data collectors in Lahore and helped to finalize wording and translation to ensure data collectors’ and respondents’ understanding of the survey items.

### 4.7. Survey Sample Size and Sampling Strategy

A generic formula was used for cross sectional surveys to assess prevalence of dispensing antibiotics without prescriptions [38]. To estimate prevalence, data from other dispensing studies in Pakistan was used, as well as data from a similar study with pharmacists in Nepal [12]. Because of the difference in context and the range in different types of respondents (e.g., physicians, nurses, pharmacists) a low-end conservative estimate of 0.20 (20%) prevalence was used. The calculation was made based on a level of confidence (z) at 0.99 and precision (d) at 0.03. The sample size estimate was calculated at 176.

Sampling was purposeful and stratified to make certain the study had representation from the various types of healthcare providers and representation from various type of private and public health facilities, pharmacies, and medicine shops. The first step of sampling was the random selection of the Lahore City Union Councils (UCs) to represent urban and rural areas. In the second step, the UCs were mapped to locate the public healthcare facilities including primary and secondary facilities with out-patient services. Private clinics and medicine shops were then selected from the areas where the public healthcare facilities were located. This process ensured that the selected private providers were serving the same communities visiting the public sector providers.

### 4.8. Survey Data Collection and Management

Data collection was conducted in September 2021. Potential survey respondents were contacted by research staff. They were informed about the study and consent was obtained prior to data collection. The quantitative survey data was collected face-to-face by trained Lahore staff using personal digital assistants (Tablets). The survey was built and managed through REDCap which is a secure website which allows investigators immediate access to data as it is being collected [39]. This enabled the investigators to monitor and address any issues in relation to data collection on a day-to-day basis. Data was downloaded from REDCap and transposed to a format compatible with SPSS statistical software. Initial data cleaning included creation of variables for knowledge and perception scales and use of descriptive statistics to screen for missing cases and outliers.

### 4.9. Survey Data Analysis

The analysis is focused on descriptive statistics and bivariate analysis to identify significant differences across types of health care providers and dependent variables including prescribing/dispensing practices, familiarity with antibiotic guidelines and policies, and knowledge and perceptions of antibiotic resistance in Pakistan. Percentages and number of respondents are presented for the descriptive data. Comparisons to determine significant differences for the bivariate analysis included Pearson’s chi square (categorical variables) and ANOVA (continuous variables). The threshold for significance was 0.05.

## Figures and Tables

**Figure 1 antibiotics-11-01418-f001:**
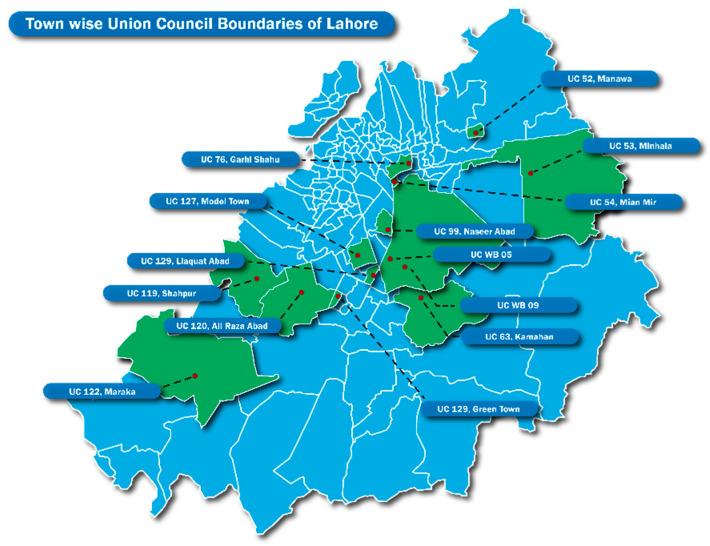
Map of Lahore with Study Union Councils (UC) Indicated.

**Table 1 antibiotics-11-01418-t001:** Demographics by current health care position.

		Physician (N = 41)	Nurse (N = 29)	LHW or LHV * (N = 10)	Midwife (N = 18)	Pharmacy Manager or Clerk (N = 49)	Medicine Shop Owner or Compounder (N = 23)	Medical Technician (N = 7)
Gender ^c^	Male	34.1% (14)	3.4% (1)	0	0	85.7% (42)	100.0% (23)	57.1% (4)
	Female	65.9% (27)	96.6% (28)	100.0% (10)	100.0% (18)	14.3% (7)	0	42.9%(3)
Age	Mean (SD)	36.4 (11.8)	30.3 (6.6)	38.7 (12.8)	32.7 (7.2)	27.7 (6.3)	34.3 (12.1)	34.7 (12.7)
Site ^c^	Rural	9.8% (4)	0	40.0% (4)	5.6% (1)	0	13.0% (3)	0
	Urban	90.2%(37)	100.0% (29)	60.0% (6)	94.4% (17)	100.0% (49)	87.0% (20)	100.0% (7)

^c^*p* < 0.001; * Lady Health Worker/Lady Health Volunteer.

**Table 2 antibiotics-11-01418-t002:** Mean ranking of common infections among patients/customers by type of health facility. (ranked from 1 [most common] to 5 [least common]).

	Private Clinics	BHUs	Pharmacies and Medicine Shops	Referral Hospitals
Upper Respiratory Infection ^a^	2.9 (1.6)	2.1 (1.3)	2.8 (1.4)	2.2 (1.1)
Gastrointestinal ^b^	2.5 (1.3)	2.6 (1.1)	1.9 (0.9)	2.2 (1.2)
Urinary Tract Infection (UTI) ^c^	2.4 (1.1)	4.0 (1.2)	3.4 (1.1)	2.9 (1.1)
Skin/Wound Infection ^c^	3.7 (1.4)	2.8 (1.3)	2.6 (1.2)	4.1 (1.1)
Pelvic Inflammatory Disease (PID) ^c^	3.7 (1.3)	3.5 (1.3)	4.5 (0.9)	3.6 (1.4)

^a^*p* < 0.05; ^b^
*p* < 0.01; ^c^
*p* < 0.001.

**Table 3 antibiotics-11-01418-t003:** Mean ranking of commonly prescribed/dispensed antibiotics on a weekly basis (ranked from 1 [most common] to 7 [least common]) by health care position.

	Physician	Nurse	LHW or LHV *	Midwife	Pharmacy Manager or Clerk	Medicine Shop owner or Compounder	Medical Technician
Amoxicillin/Augmentin ^b^	1.7 (SD 1.3)	2.7 (SD 1.6)	1.6 (SD 0.8)	1.9 (SD 1.0)	3.0 (SD 1.5)	2.5 (SD 1.8)	1.7 (SD 1.4)
Ampicillin ^c^	4.0 (SD 4.3)	2.6 (SD 2.8)	1.7 (SD 2.5)	2.0 (SD 2.7)	5.3 (SD 1.4)	4.0 (SD 2.9)	3.4 (SD 2.6)
Azithromycin ^b^	3.7 (SD 2.3)	2.8 (SD 2.3)	1.9 (SD 2.6)	1.7 (SD 2.3)	2.2 (SD 1.3)	1.8 (SD 1.5)	2.3 (SD 2.6)
Cefixime ^a^	2.4 (SD 1.6)	2.4 (SD 1.9)	1.5 (SD 1.8)	1.3 (SD 1.6)	2.6 (SD 1.4)	3.0 (SD 1.1)	3.4 (SD 2.0)
Ceftriaxone (injectable) ^c^	3.3 (SD 2.4)	1.6 (SD 1.3)	1.6 (SD 2.7)	1.4 (SD 1.7)	5.7 (SD 1.8)	5.1 (SD 1.7)	3.6 (SD 2.6)
Ciprofloxacin ^c^	3.3 (SD 1.6)	2.5 (SD 1.4)	1.2 (SD 0.9)	2.1 (SD 1.5)	3.2 (SD 1.4)	3.0 (SD 1.6)	3.3 (SD 2.0)
Co-trimoxazole ^c^	4.1 (SD 2.7)	3.3 (SD 3.3)	2.5 (SD 1.8)	2.0 (SD 2.7)	5.5 (SD 1.6)	4.8 (SD 1.9)	4.4 (SD 2.8)

^a^*p* < 0.05; ^b^
*p* < 0.01; ^c^
*p* < 0.001. * Lady Health Worker/Lady Health Volunteer.

**Table 4 antibiotics-11-01418-t004:** Likelihood of prescribing/dispensing antibiotics for children with specific symptoms (likely/very likely) by health care position.

	Physician	Nurse	LHW or LHV *	Midwife	Pharmacy Manager or Clerk	Medicine Shop Owner or Compounder	Medical Technician
Fever ^a^	48.8% (20)	37.9% (11)	30.0% (3)	44.4% (8)	21.7% (10)	21.7% (5)	0
Cough	26.8% (11)	41.4% (4)	40.0% (4)	33.3% (6)	26.5% (13)	34.8% (8)	0
Sore Throat ^a^	75.6% (31)	44.8% (13)	60.0% (6)	52.9% (9)	47.9% (23)	43.5% (10)	42.9% (3)
Diarrhea ^b^ (lasting more than one day)	51.2% (21)	65.5% (19)	50.0% (5)	61.1% (11)	43.8% (21)	30.4% (7)	28.6% (2)

^a^*p* < 0.05; ^b^
*p* < 0.01. * Lady Health Worker/Lady Health Volunteer.

**Table 5 antibiotics-11-01418-t005:** Likelihood of prescribing/dispensing antibiotics for an adult with specific symptoms (likely/very likely) by health care position.

	Physician	Nurse	LHW or LHV *	Midwife	Pharmacy Manager or Clerk	Medicine Shop Owner or Compounder	Medica Technician
Fever	48.8% (20)	44.8% (13)	60.0% (6)	55.6% (10)	36.7% (18)	60.9% (14)	42.9% (3)
Cough	31.7% (13)	41.4% (12)	40.0% (4)	55.6% (10)	51.0% (25)	43.5% (10)	28.6% (2)
Sore Throat	70.0% (28)	58.6% (17)	80.0% (8)	55.6% (10)	77.6% (38)	65.2% (15)	28.6% (2)
Diarrhea ^b^ (lasting more than one day)	58.5% (24)	55.2% (16)	80.0% (8)	61.1% (11)	70.8% (34)	39.1% (9)	42.9% (3)
Burning sensation when urinating	63.4% (28)	55.2% (16)	90.0% (9)	61.1% (11)	43.8% (21)	56.5% (13)	28.6% (2)

^b^*p* < 0.01. * Lady Health Worker/Lady Health Volunteer.

**Table 6 antibiotics-11-01418-t006:** Likelihood of prescribing/dispensing antibiotics for wounds (likely/very likely) by health care position.

	Physician	Nurse	LHW or LHV *	Midwife	Pharmacy manager or Clerk	Medicine Shop owner or Compounder	Medical Technician
Redness and/or swelling around a wound ^b^	85.4% (35)	62.1% (18)	100.0% (10)	88.9% (16)	58.3% (28)	65.2% (15)	28.6% (2)
Discharge from wound ^b^	87.8% (36)	69.0% (20)	100.0% (10)	94.4% (17)	79.2% (38)	69.6% (16)	14.3% (1)

^b^*p* < 0.01. * Lady Health Worker/Lady Health Volunteer.

**Table 7 antibiotics-11-01418-t007:** Mean scores for knowledge and perceptions of AMR (severity, self-efficacy, response efficacy, and response cost) by health care position.

	Physician	Nurse	LHW or LHV *	Midwife	Pharmacy Manager or Clerk	Medicine Shop owner or Compounder	Medical Technician
Knowledge ^c^ Possible range 0 to 12	7.4 (SD 1.7)	8.4 (SD 1.7)	8.7 (SD 1.6)	8.7 (SD 1.5)	9.2 (SD 1.7)	8.4 (SD 1.5)	8.0 (SD 2.3)
Severity ^c^ Possible range 4 to 16	13.1 (SD 1.8)	11.2 (SD 1.8)	11.9 (SD 1.3)	11.7 (SD 1.2)	11.8 (SD 1.5)	11.8 (SD 1.4)	11.3 (SD 1.0)
Self-efficacy ^c^ Possible range 3 to 12	9.8 (SD 1.1)	8.7 (SD 1.2)	8.5 (SD 1.3)	9.3 (SD 1.0)	9.0 (SD 1.2)	8.7 (SD 1.1)	8.0 (SD 1.0)
Response efficacy Possible range 2 to 8	5.8 (SD 0.8)	5.2 (SD 0.8)	5.4 (SD 0.7)	5.6 (SD 0.8)	5.4 (SD 1.0)	5.4 (SD 0.7)	5.4 (SD 1.3)
Response cost ^c^ Possible range 2 to 8	4.3 (SD 1.1)	4.1 (SD 0.9)	4.4 (SD (1.3)	4.8 (SD 1.0)	5.2 (SD 1.1)	5.0 (SD 1.3)	4.1 (SD 1.1)

^c^*p* < 0.001. * Lady Health Worker/Lady Health Volunteer.

**Table 8 antibiotics-11-01418-t008:** Knowledge and Perception Scoring and Items.

**Knowledge** **(Range 0 to 12)** **Response Options: True/False**	Antibiotics have saved millions of lives
	Antibiotics are good for treating infections caused by viruses
	Antibiotic kill bacteria that cause illness
	Antibiotics kill good bacteria that protect the body from infection
	Antibiotics can cure colds and flu
	Antibiotics can be used to treat COVID-19
	It is safe to use antibiotics from family, friends and others
	Some people have allergies to antibiotics
	A person should only stop using an antibiotic after consulting the prescriber (physician)
	Antibiotic resistance is not a concern because new antibiotics will be available in the future
	Antibiotic resistance is a significant problem in Pakistan
	The majority of antibiotic use occurs in inpatient hospital settings in Pakistan
**Severity of AMR** **(Range 4 to 16)** **Response Options:** **Strongly Agree, Agree, Disagree, Strongly Disagree**	Antibiotic resistance affects my patients’/customers’ health and well-being
	Antibiotic resistance affects my ability to help my patients’/customers’ recover from infectious diseases
	Antibiotic resistance increases the cost of health care
	Antibiotic resistance could affect my family’s health and well-being
**Self-efficacy to Make Changes** **(Range 3 to 12)** **Response Options:** **Strongly Agree, Agree, Disagree, Strongly Disagree**	I can change my antibiotic prescribing/dispensing practices based on government guidelines
	I can explain to my patients/customers why they do NOT need an antibiotic in certain situations (e.g., viral infections
	I can be an advocate for antibiotic stewardship with my peers and colleagues
**Response-Efficacy** **(Range 2 to 8)** **Response Options:** **Strongly Agree, Agree, Disagree, Strongly Disagree**	I can contribute to decreasing antibiotic resistance by changing my prescribing/dispensing practices
	The few antibiotics that I prescribe/dispense do not affect antibiotic resistance in Pakistan
**Response Costs** **(Range 2 to 8)** **Response Options:** **Strongly Agree, Agree, Disagree, Strongly Disagree**	I will lose customers or patients if I decrease prescribing/dispensing antibiotics
	I would be considered an irresponsible healthcare provider if I did not provide antibiotics to patients/customers when they request them

## Data Availability

The data presented in this study are available on request from the corresponding author. The data are not publicly available as this was not stipulated within the written consents for this study.

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
