# Peer review of "A Formative Assessment of Antibiotic Dispensing/Prescribing Practices and Knowledge and Perceptions of Antimicrobial Resistance (AMR) among Healthcare Workers in Lahore Pakistan"

_antibiotics, 2022, doi:10.3390/antibiotics11101418_

Round 1

Reviewer 1 Report

This manuscript entitled with "A Formative Assessment of Antibiotic Dispensing/Prescribing Practices and Knowledge and Perceptions of Antimicroial Resistance (AMR) among Healthcare Workers in Lahore Pakistan" conducted a cross-sectional survey from private and public health facilities in 14 union councils, Lahore Pakistan. This manuscript was well thought and well-organized. It can be accepted for publication if the following concerns can be addressed:

Major:

  1. Discussion part: it is good to study the effect of COVID-19 on AMR, however, this section lack a broad and systematic comparison to other countries. Please include that.
  2. Research site: please show the locations in a map.

Minor:

  1. Line 128: the full name of BHU should be included here, not line 307.
  2. Line 149: "(p=.66)" should be "(p=0.66)"; same issues for lines 348 and 349.
  3. Table 2: Typo for the 3rd line "Gastro-intestinal bl"?
  4. Line 190: Explanation for "WHO AWaRe"?
  5. Line 220: "(SD(1.2)" should be "(SD 1.2)".

Author Response

Reviewer 1

This manuscript entitled with "A Formative Assessment of Antibiotic Dispensing/Prescribing Practices and Knowledge and Perceptions of Antimicrobial Resistance (AMR) among Healthcare Workers in Lahore Pakistan" conducted a cross-sectional survey from private and public health facilities in 14 union councils, Lahore Pakistan. This manuscript was well thought and well-organized. It can be accepted for publication if the following concerns can be addressed:

Major:

  1. Discussion part: it is good to study the effect of COVID-19 on AMR, however, this section lack a broad and systematic comparison to other countries. Please include that. We have included additional information and references regarding COVID 19 and AMR [lines 229 to 248]
  2. Research site: please show the locations in a map. Map has been included.

Minor:

  1. Line 128: the full name of BHU should be included here, not line 307. Corrected
  2. Line 149: "(p=.66)" should be "(p=0.66)"; same issues for lines 348 and 349. Corrected
  3. Table 2: Typo for the 3rd line "Gastro-intestinal bl"? Corrected
  4. Line 190: Explanation for "WHO AWaRe"? Added information
  5. Line 220: "(SD(1.2)" should be "(SD 1.2)". Corrected

Reviewer 2 Report

Overall interesting study regarding the knowledge gap among prescribers of antibiotics in a specific geographical region. 

Please describe the survey design and validation process. 

Please describe what statistical tests were used for comparison in the methods section and what the threshold for significance was. 

Page 5 - upon reading these results, my concern is that they are relatively non-specific and that the authors should refine their focus to appropriate usage criteria for antibiotics that is more specific than general concepts such as fever and diarrhea. 

Table 1- I would suggest adding a total N for each major profession category at the top of the table. (eg how many total physicians responded) 

Author Response

Overall interesting study regarding the knowledge gap among prescribers of antibiotics in a specific geographical region. 

Please describe the survey design and validation process.  Information added lines 370 to 379. 

Sampling was purposeful and stratified to make certain the study had representation from the various types of healthcare providers and representation from various type of private and public health facilities, pharmacies, and medicine shops. The first step of sampling was the random selection of the Lahore City Union Councils (UCs) to represent urban and rural areas. In the second step, the UCs were mapped to locate the public healthcare facilities including primary and secondary facilities with out-patient services. Private clinics and medicine shops were then selected from the areas where the public healthcare facilities were located. This process ensured that the selected private providers were serving the same communities visiting the public sector providers

Please describe what statistical tests were used for comparison in the methods section and what the threshold for significance was. Information added line 395 to 398.

Comparisons to determine significant differences for the bivariate analysis included Pearson’s chi square (categorical variables) and ANOVA (continuous variables). The threshold for significance was 0.05. 

Page 5 - upon reading these results, my concern is that they are relatively non-specific and that the authors should refine their focus to appropriate usage criteria for antibiotics that is more specific than general concepts such as fever and diarrhea.

The data was collected using a validated tool.  This tool was used to collect data in Nepal and a qualitative study was conducted in Pakistan to identify/confirm common ailments presenting and antibiotics prescribed.  In Pakistan as in many LMIC, antibiotics are prescribed or dispenses based on patient presentation of symptoms (e.g., cough, fever) to pharmacists, medicine shop owners, and healthcare providers.   

Table 1- I would suggest adding a total N for each major profession category at the top of the table. (eg how many total physicians responded)  Information added.

Reviewer 3 Report

Rakhshani et al., have reported the manuscript entitled as “A Formative Assessment of Antibiotic Dispensing/Prescribing Practices and Knowledge and Perceptions of Antimicrobial Resistance (AMR) among Healthcare Workers in Lahore Pakistan”. The ms domain is very relevant and interesting. The objective of this paper was to describe antibiotic dispensing/prescribing practices and underlying factors associated with these practices among community-based healthcare workers.

For this manuscript, my very 1st observation is the interesting fact that the field work / study area is Lahore that is quite away from the investigators i.e. Karachi, Pakistan and Michigan, USA. Authors should better describe their data collection procedure, to justify this far distance data acquisition from the field.

As per the manuscript, estimated population of study area (Lahore) is 13 million, while the region is divided into ten zones/towns, and each town is subdivided into union councils (UC). These UCs were selected based on geographic, socio-cultural, health, and economic factors including: 1) resident ethnicities; 2) population density; 3) mean household income and primary occupations; 4) migration status (rural-to-urban); 5) types of health services available including both public and private facilities. Prior to data collection, the UCs were mapped for locations of healthcare facilities and pharmacies and medicine shops. In this context, only 177 respondents (including various types i.e. physicians, nurses, lady health workers/volunteers, midwives, pharmacy and medicine shop employees, and medical technicians) seem not the real representation of the sampled area. Also, the authors have mentioned the presence of different ethnic groups in the sampling area, but no such data is presented in the document.

Moreover, as per section 4.2 of the manuscript, many public healthcare facilities (classified into five categories) in Lahore provide heavily subsidized healthcare to the communities. Lahore also has a large private healthcare sector consisting of stand-alone clinics and tertiary level hospitals. This information raises two queries i.e. (1) how likely is the un-prescribed medicines ‘practice’ in such healthcare system? (2) From all these healthcare facilities, would it not be better to collect the data from patient files and other official (primary) documented sources rather than the interviews/survey.

The bivariate analyses of the data in this study can better be presented in some figure/diagram, as the whole ms data in tabular form is monotonous.

At the end, the survey-questionnaire and interview details should also be shared with journal – as per publisher’s instructions.

Author Response

Rakhshani et al., have reported the manuscript entitled as “A Formative Assessment of Antibiotic Dispensing/Prescribing Practices and Knowledge and Perceptions of Antimicrobial Resistance (AMR) among Healthcare Workers in Lahore Pakistan”. The ms domain is very relevant and interesting. The objective of this paper was to describe antibiotic dispensing/prescribing practices and underlying factors associated with these practices among community-based healthcare workers.

 For this manuscript, my very 1st observation is the interesting fact that the field work / study area is Lahore that is quite away from the investigators i.e. Karachi, Pakistan and Michigan, USA. Authors should better describe their data collection procedure, to justify this far distance data acquisition from the field.

PHC Global has three offices across Pakistan. Its head office is in Karachi, but has executive and technical staff based in Lahore and Islamabad. The Lahore site was selected by PHC Global as they Pubjab Primary and Secondary Health Department is very supportive of research efforts regarding AMR and stewardship. The co-principal investigator who conducted the field data collection and analysis has been trained and worked in Pakistan as a clinician and public health practitioner.  This co-PI has conducted research studies in three provinces of the country and is well versed in the healthcare system in Pubjab. The Michigan-based PI has extensive experience working on AMR and stewardship and infectious disease in South Asia including Pakistan. We have also added the following sentence to clarify that our research team was in Lahore (lines 383 and 384).

The quantitative survey data was collected face-to-face by trained Lahore staff using personal digital assistants (Tablets).

As per the manuscript, estimated population of study area (Lahore) is 13 million, while the region is divided into ten zones/towns, and each town is subdivided into union councils (UC). These UCs were selected based on geographic, socio-cultural, health, and economic factors including: 1) resident ethnicities; 2) population density; 3) mean household income and primary occupations; 4) migration status (rural-to-urban); 5) types of health services available including both public and private facilities. Prior to data collection, the UCs were mapped for locations of healthcare facilities and pharmacies and medicine shops. In this context, only 177 respondents (including various types i.e. physicians, nurses, lady health workers/volunteers, midwives, pharmacy and medicine shop employees, and medical technicians) seem not the real representation of the sampled area. Also, the authors have mentioned the presence of different ethnic groups in the sampling area, but no such data is presented in the document.

The variations included in selecting the UCs was to have differences in clients/patients seen by the healthcare providers. We do not have specific information about health facility clients/patients.   Only healthcare providers were interviewed.  While there is variation in ethnicity within Lahore and the selected UCs, all respondents were of the same ethnicity. Languages spoken included Punjab, Urdu, and English.  We have modified lines 317 to 323. 

The current research was conducted in 14 UCs in Lahore and all respondents were of the same ethnic background. These UCs were selected based on geographic, socio-cultural, health, and economic factors including: 1) population density; 2) mean household income and primary occupations; 3) migration status (rural-to-urban);  and, 4) types of health services available including both public and private facilities.

Moreover, as per section 4.2 of the manuscript, many public healthcare facilities (classified into five categories) in Lahore provide heavily subsidized healthcare to the communities. Lahore also has a large private healthcare sector consisting of stand-alone clinics and tertiary level hospitals. This information raises two queries i.e. (1) how likely is the un-prescribed medicines ‘practice’ in such healthcare system? (2) From all these healthcare facilities, would it not be better to collect the data from patient files and other official (primary) documented sources rather than the interviews/survey.

Dispensing of medications are common across Pakistan.  The data speak for itself – there was significant dispensing of medications from the study population.  In Pakistan, patient records are maintained only in tertiary care hospitals which are almost all paper based and mostly incomplete. 

The bivariate analyses of the data in this study can better be presented in some figure/diagram, as the whole ms data in tabular form is monotonous. We feel that the data are better presented in tables – the data can be more easily read and data can be compared across the tables. 

At the end, the survey-questionnaire and interview details should also be shared with journal – as per publisher’s instructions. We have added the final survey to the end of the document as an appendix.

Round 2

Reviewer 1 Report

The authors revised the manuscript accordingly, however, the map should establish the locations of healthcare facilities that were investigated in Lahore, not the locations of Lahore in Parkistan. 

Author Response

We have submitted a version with a map which shows the location of the 14 UCs where data were collected. We feel that showing specific health centers could compromise confidentiality for the respondents - many of the health centers have only a few healthcare providers.

Reviewer 2 Report

Thank you for making appropriate revisions 

Author Response

No additional comments from this reviewer.

Reviewer 3 Report

The authors should extend the legend for Figure 1, for better elaboration e.g. they may say that the red triangle indicates the study area in Punjab, Pakistan with x, y coordinates.

The sample size in this study i.e. only 176/177 respondents (including various types i.e. physicians, nurses, lady health workers/volunteers, midwives, pharmacy and medicine shop employees, and medical technicians) seems not realistic for the actual representation of the sampled area; particularly for the publication in journal like ‘Antibiotics’. If the authors feel otherwise they may better cite some comparable studies with such sample sizes. Several demographic studies report 10% of the study-population as their realistically good sample size.

Author Response

We have already outlined our sample size calculation and strategy. However, we have not included a 'limitations' section to address concerns regarding study sampling (see below for content) 

The current study was conducted with 178 health care workers in 14 Union Councils within Lahore. The size of the study may limit generalization of the findings for elsewhere in Lahore and in Pakistan. However, our findings are consistent with and supportive of similar research in Pakistan and the region. The data on antibiotic prescribing and dispensing was subjective and based on reported practices.  Further research is needed utilizing more objective methodologies including standardized patients and implementation of stringent antibiotic consumption and use protocols in healthcare facilities. The current data provide an overview of the magnitude of sub-optimal antibiotic prescribing and dispensing practices and potential areas of attention for targeted stewardship interventions.